# RC-LAHR: Road-Side-Unit-Assisted Cloud-Based Location-Aware Hybrid Routing for Software-Defined Vehicular Ad Hoc Networks

**DOI:** 10.3390/s24041045

**Published:** 2024-02-06

**Authors:** Manish Kumar, Ram Shringar Raw

**Affiliations:** 1Department of Computer Science and Engineering, University School of Information, Communication and Technology, Guru Gobind Singh Indraprastha University, Dwarka, New Delhi 110078, India; manishk19071984@gmail.com; 2Department of Computer Science and Engineering, Netaji Subhas University of Technology, East Campus, Delhi 110031, India

**Keywords:** cloud-based VANET, SDN, SD-VANET, software-defined networking, VANET

## Abstract

The reliability of the communication link is quite common and challenging to handle as the topology changes frequently in vehicular ad hoc networks (VANETs). Another problem with VANETs is that the vehicles are from different manufacturers. Hence, the heterogeneity of hardware is obvious. These heterogeneity and reliability problems affect the message dissemination in VANETs. This paper aims to address these challenges by proposing a robust routing protocol capable of ensuring reliable, scalable, and heterogeneity-tolerant message dissemination in VANETs. We first introduced a hybrid hierarchical architecture based on software-defined networking (SDN) principles for VANETs, leveraging SDN’s inherent scalability and adaptability to heterogeneity. Further, a road-side unit (RSU)-assisted cloud-based location-aware hybrid routing for software-defined VANETs (SD-VANETs) that we call RC-LAHR was proposed. RC-LAHR was rigorously tested and analyzed for its performance in terms of packet delivery ratio (PDR) and end-to-end delay (EED), along with a comprehensive assessment of network traffic and load impacts on cloud infrastructure and RSUs. The routing protocol is compared with state-of-the-art protocols, Greedy Perimeter Stateless Routing (GPSR) and Opportunistic and Position-Based Routing (OPBR). The proposed routing protocol performs well as compared to GPSR and OPBR. The result shows that the EED is reduced to 20% and the PDR is increased to 30%. The network reliability is also increased up to 5% as compared to the OPBR and GPSR.

## 1. Introduction

VANET is a network of intermittently connected vehicles. In VANETs, the nodes are highly mobile in nature. The extreme mobility of cars and frequent topology changes distinguish vehicular ad hoc networks from other mobile ad hoc networks. Hence, it is very challenging to disseminate the message to the nodes with high reliability. To cope with the message dissemination challenge, many routing protocols have been developed [1,2,3,4,5].

In VANETs, location-based routing protocols [6,7,8,9,10], which are also known as position-based routing protocols and topology-based routing protocols, have been frequently used. Topology-based routing is not more effective since the routes determined by it are ineffective for transmitting data in highly dynamic environments [11]. Another crucial need for location-based routing is its scalability, which implies that the routing protocol must be designed in a way that prevents performance from being impacted by changes in vehicle density. This is only possible if the protocol can use localized information and the choice made by the cars is only based on the information available in their area [12]. For the reasons listed above, location-based routing protocols are more common than others to provide sufficient performance in line with VANET features [13,14].

Traditional VANETs employ a conventional network architecture where each node serves as a router, combining forwarding and routing functions into a single unit [15,16,17,18,19]. Nevertheless, this conventional approach has notable drawbacks, such as its inability to accommodate heterogeneity, scalability issues, limited programmability, and other concerns [20]. SDN [21] has emerged as a cutting-edge solution to tackle the challenges faced by traditional routers. In the SDN approach, the routing module is distinct from the router itself and is controlled by a separate control plane. Whenever routing is required, the control plane sets rules for the forwarding module. This clear separation of functions enables more flexible and effective network management. SD-VANETs are becoming more popular among researchers. The concept of SDN may be utilized to solve challenges such as heterogeneity and scalability. The control planes handle routing decisions in SDN. The control plane for the SDN implementation in VANETs could utilize a cloud-based solution. Using the rapid processing and infinite storage capabilities of the cloud, the network design may potentially be managed and maintained [22].

This research is aimed at providing an SDN-based VANET architecture to solve the problems caused by heterogeneity and scalability. RC-LAHR, a cloud-based location-aware hybrid routing protocol, is also proposed for the architecture in question to find the suitable route and disseminate the data.

The organization of this paper is as follows. In Section 2, the background and related work are discussed. The motivation and research contributions are described in Section 3. Section 4 describes the system model and proposed framework. In Section 5, the proposed routing protocol is discussed in detail. In Section 6, the computational complexity of the proposed routing algorithm is proven. Section 7 provides the detailed result analysis of the proposed routing algorithm. Section 8 provides the discussion of the results. The conclusion is provided in Section 9. Finally, the limitations and future work are discussed in Section 10.

## 2. Background and Related Work

VANETs exhibit notable distinctions when compared to traditional wireless networking methodologies. The high degree of mobility that its nodes exhibit is what makes VANETs unique. The conventional network routing protocol is not directly applicable to VANETs. In recent years, researchers have put forth a range of routing protocols. However, the routing mechanism in an SDN-based VANET is similar to that in conventional VANETs, but it necessitates certain modifications to align with the SDN architecture. Significant contributions have been made in recent scholarly publications in the area of routing in SD-VANETs. 

One of the most popular location-based routing protocols used in VANETs is GPSR [23]. It uses a greedy forwarding strategy that relies only on the information of nearby nodes in the network’s architectural context. When greedy forwarding is not feasible, the algorithm compensates by redirecting the packet around the periphery of the region that is beyond the immediate area. GPSR outperforms shortest path and ad hoc routing protocols in terms of scalability. GPSR needs to store information about the immediate network structure, which enables it to handle a growing number of network destinations without difficulty. The GPS efficiently utilizes local topology information to rapidly identify accurate new routes within the rapid-changing mobility topology. The GPSR protocol depends on the fact that forwarding choices in geographic routing relies exclusively on information about the neighbor. Despite the several advantages of GPSR and its use of local information for routing decisions, it also has notable limitations. In sparse network conditions, where nodes are few and far away from each other, GPSR may struggle to find a suitable next hop, leading to routing inefficiencies.

A recent study [24] presents a novel location-based routing protocol, OPBR, for VANETs that combines opportunistic and location-based approaches. This protocol considers variables such as the quality of the links, the density of the nodes, and the location of the nodes to determine the route. A greedy forwarding mechanism is utilized by the system, and ideal candidate nodes are selected through a combination of opportunistic and location-based strategies. In addition to this, it identifies the nodes that have reached the end of their allotted time in the routing process and removes them from the running order. While OPBR protocols offer advantages like efficiency in dynamic networks, reduced overhead, and scalability, they also face challenges such as dependence on accurate location data, performance issues in sparse networks, potential delays, risks of routing loops, and the lack of inherent support for varying QoS requirements.

Some researchers have also investigated routing in SD-VANETs. In their study, the authors of [25] introduced an innovative packet routing scheme for SD-VANETs. This scheme incorporates a flow instantiation operation based on source routing. The routing problem was transformed into an integer linear programming problem. A novel approach involving incremental packet allocation was proposed as a solution to address the routing problem, with the aim of reducing time complexity. Although the scheme reduced the time complexity of routing, the VANET architecture lacks flexibility that makes large-scale service and protocol deployment difficult. In a scholarly article, the authors of [26] presented a routing framework that utilizes SDN to facilitate message propagation within VANETs. The routing protocol that was proposed computes routing paths that are globally optimized, thereby reducing the routing overhead. In a recent study, the authors of [27] introduced a novel routing protocol that operates based on the principles of spray and pray multiple-copy routing. The spray technique was employed for the purpose of elimination, while pray was utilized to mitigate the delay in packet delivery. In the context of VANETs, a protocol known as a GeoBroadcast was proposed by the authors of [28] for implementation within the SDN architecture. Within this protocol, the transmitting node is responsible for dispatching a recurring warning message to apprise the receiving node. To facilitate the effective transmission and delivery of warning messages to the intended recipients, the Floodlight SDN controller was implemented with RSU location management and GeoBroadcast components. In their study, the authors of [29] put forth a routing protocol specifically designed for cognitive radio software-defined vehicular networks. The protocol consists of two phases, namely, the registration phase and the route prediction phase. The RSU served as the local controller for the registration phase, while the main controller was responsible for the route prediction phase. In their study, the authors of [30] put forward a cross-layer routing approach. In addition to channel allocation and link duration, the routing protocol incorporates the metric of vehicles’ location and velocity. The vehicle initiates the registration process by transmitting a hello message to the local controller. In [31], the authors proposed a routing protocol for network state management that operates based on the concept of lifetime. The utilization of 4G cellular technology was employed for communication purposes. The protocol initially verifies the existence of a V2V routing path from the source to the destination. Security is one of the major concerns in VANET routing. The secure routing was proposed in [31,32,33,34,35,36,37,38], which provided security in VANET routing. Other methodologies like artificial intelligence and machine learning [39,40,41] for different cases [42,43,44] are also used in the literature.

While the routing protocols discussed above have their unique strengths in certain aspects of VANET communication, their limitations in sparse networks, dependence on accurate location data, and the lack of QoS support highlight the need for more adaptable and flexible network architectures. Moreover, the application of SDN in VANETs presents its own set of challenges, necessitating innovative solutions to fully harness its potential in these highly dynamic environments.

## 3. Motivation and Research Contribution

The available research on VANETs lacks advancements in link reliability, especially in the context of software-defined networking. The primary motivation of this work is to exploit the flexibility and adaptability of SDN architecture to maximize link reliability and, hence, the packet delivery ratio. The enhancements in wireless technology, like LTE and high-speed Wi-Fi technology, make it easier to separate data transmission from traffic control transmission. Through GPS technology, it is easier to keep track of the global view of the network. Other researchers have implemented SDN in VANETs, but these approaches lack the full potential of SDN as they use a single interface to find the route and data dissemination. Also, the work carried out with VANETs has not paid attention to the heterogeneity of the vehicle hardware. The above discussion motivated us to carry out this research.

### Research Contribution

This article makes the following contributions:A novel hybrid hierarchical architecture within the SDN framework for VANETs is introduced that integrates cloud computing capabilities with VANET infrastructure, addressing the dynamic nature and heterogeneity of vehicular networks.The formulation of the RC-LAHR protocol is a primary contribution. This protocol leverages cloud computing for data handling and location-based information to optimize routing.RC-LAHR significantly improves the reliability and scalability of message dissemination in VANETs. By efficiently handling frequent topology changes and varied hardware, it ensures more stable and consistent communication in vehicular networks.A time complexity analysis is performed to test the efficacy of the proposed routing protocol. The proposed routing protocol works in *O*(*n*^3^) time complexity.Our comprehensive comparative analysis of RC-LAHR with existing protocols like GPSR and OPBR, focusing on PDR, EED, and network reliability, provides empirical evidence of the superiority of RC-LAHR.

## 4. System Model and Proposed Framework

This section first provides the notations and abbreviations used in this work. Further, it describes the system model and proposed framework. 

### 4.1. Proposed Architecture

In this research, a two-lane traffic scenario was considered that resembles real-time traffic. The RSU was placed in the middle of the road to cover all portions of the road. A hybrid hierarchical model was used, which can maintain load balance even in dense traffic. Each RSU can handle the control plane request. If the RSU is not able to handle the request in a situation where the route is not available or the destination node is not known, it sends the request to control plane layer 2. The two network interfaces are LTE for RSU to vehicle (R2V) communication and high-speed low-range Wi-Fi for vehicle to vehicle (V2V) communication. The cloud server was implemented at the top of all layers. It stores all the software updates, the logical view of global topology, and control policies. It also facilitates the RSUs. If the RSU is not able to handle the request, it sends the request to the cloud server. The cloud server has an overall topological view. Therefore, it responds to the request of the RSU, and the RSU further disseminates it to the nodes as per the request. All RSUs are connected to a high-speed wired network to communicate with each other. Figure 1 illustrates the system model broadly.

The proposed framework can be divided into three planes.
Data Plane: This plane of the framework contains the data-forwarding module. The vehicles are equipped with high-speed Wi-Fi to forward the data, based on IEEE 802.11ac standards [45]. However, like the normal SDN, these data planes also have the capability to store the route locally for a given time period. If the sender node starts sending data, it first searches the locally stored routes. If the route is valid, it starts transmitting data; otherwise, it goes to the control plane for the new rules.Control Plane: The control plane is divided into two parts to reduce the load on the control plane. Control plane layer 1 covers the logical local topology of the network. RSU serves as layer 1 of the control plane. Layer 2 collects all the information from the RSUs in its region, and finally, it keeps the global topological view of its region. It may communicate with other control plane layer 2 stations. Control plane layer 2 is also capable of performing analyses and making decisions. This layer can be used for machine learning and AI-based utilities in SD-VANETs. The cloud server has been used to act as a control plane layer 2.Application Plane: The application plane of the framework provides facilities like network security policies, user-based applications, the internet, etc. Control plane layer 2 interacts with the application plane using the northbound API of the OpenFlow protocol [46].

## 5. RC-LAHR: RSU-Assisted Cloud-Based Location-Aware Hybrid Routing

In this section, the cloud-based routing protocol for the proposed SD-VANET architecture is presented. All scenarios to find the destination node are discussed in detail. Further, the workings of routing are discussed. Figure 2 illustrates all the possibilities of finding the destination node. The RSU was placed on the divider of the road to obtain maximum coverage.

As shown in Figure 2, there are four possibilities for finding the destination node. In this figure, A is the sender node and B, C, D are the possible destination nodes. According to these possible nodes, the following four types of scenarios are possible. 


The destination node is found in the range of the sender vehicle: Suppose that the sender node wants to communicate with the destination node, as shown in Figure 3.


The sender node broadcasts the “hello” control message to all the vehicles in its range. If the destination node is available in the range, the destination node will send a reply. Figure 4 shows the communication process.


2.The destination node is found in the range of the RSU but outside the range of the sender: The destination node is outside the range of the sender node, but the destination node is inside the range of the same RSU in which the sender node is available, as shown in Figure 5.


In this case, the sender node sends the route request to the RSU. The RSU checks its routing table. The RSU sends the route information to all the intermediate nodes, the sender node, and the destination node. Figure 6 illustrates the communication process.


3.The destination node is outside the range of the parent RSU but in the range of a neighboring RSU: In this case, the destination node is outside the range of the parent RSU, but inside the range of a neighboring RSU, as illustrated in Figure 7.


If the parent RSU fails to get the route itself, then it broadcasts the route request to all its neighboring RSUs to find the route. Figure 8 shows the detailed communication process.


4.The destination is not in the range of neighboring RSUs: If neither the parent RSU nor neighboring RSUs get the route for further communication, the parent RSU forwards the route request to the cloud as it has overall network information to find routes. Figure 9 illustrates the representation of the scenario. Figure 10 displays the communication process.


All of these situations use the RSU as a control plane layer 1 part. Its job is to store data about vehicles in its coverage area so that it can keep a localized view of the network topology. If both the sender and destination nodes fall within the coverage area of the same RSU, the RSU will undertake the task of route calculation. Subsequently, the calculated route will be transmitted to the source node, the destination node, and any intermediate nodes that exist along the route.

The installation of a high-speed cloud server in control plane layer 2 facilitates the containment of all information on RSUs, specifically the global view of the topology. When the sender requires the route to a specific destination node and the RSU determines that the destination is not listed in its routing table, it initiates a request to the server to locate the appropriate route. The server maintains a comprehensive understanding of the network’s topology, allowing it to compute the optimal route. This information is then disseminated to all RSUs, which subsequently update their own routing tables and transmit the route to the relevant nodes. Figure 11 illustrates the details of the proposed routing protocol. The different phases of the proposed routing scheme are discussed in the following section.

### 5.1. Registration Phase

When a vehicle enters the RSU range, it sends the join packet to RSU, and RSU makes the entry in its table. The newly entered vehicle sends the hello packet to all vehicles in its range and receives a reply from its neighbors. The vehicle makes an entry or updates its local routing table. If any change occurs in its local routing table, the node informs the RSU about the change. This process is repeated by all vehicles at certain intervals.

The RSU maintains a table in which it keeps the entries of all neighboring nodes of the vehicles registered with it. The RSU updates the entry on the main server. The cloud-based control plane layer 2 also maintains the table in which all information sent by the RSU is kept. Hence, it maintains the global view of the topology. The registration phase uses Algorithm 1 for registration.


**Algorithm 1:** Registration**Input**: VID=Vehicle ID, Pos=Current Location**Output**: Vehicle registration1. ***Begin***2.   *RSUID*
←
* getRSUID(VID, Pos)*
3.   ***If** Ischanged(RSUID)==True **then***4.    *List_neighbour*
←
*Hello_broadcast (VID, Pos)*
5.     *UpdateRSU(VID, List_neighbour)*6.    ***Else***7.    ***If** isNeighbourChanged(List_neighbour) **then***8.     *updateListNeighbour()*9.     *updateRSU(VID, List_neighbour, Pos)*10.    ***End If***11.    ***End If***12.   ***End***


### 5.2. Route Discovery

Whenever a node needs to send data to another node, first, it checks whether the node is in its neighbor list. If it is present in the list, it just receives confirmation from the destination node, and after that, the node starts sending the data packets to the destination node. If the sender does not find the node in its table, it sends the route request to the RSU. The RSU checks if the destination node is inside its range. If the destination node is in range, it just calculates the route using Find Route Algo and sends RREP to all the nodes in the range.

If the destination node is not in the range of the RSU, it sends the route request to the neighboring RSUs. If the RSU does not find the route in its routing table, it sends the route request to its neighboring RSUs. If the neighboring RSU does not find the route, then finally the route request is sent to the cloud, which acts as control plane layer 2. The cloud server stores the global view of the topology. Control plane layer 2 finds the route using the Find Route algorithm, and it sends the route to all concerned RSUs from which the node is picked. Further, all RSUs send the RREP to all the nodes in the range. Algorithm 2 is described as the routing protocol.
**Algorithm 2:** FindRoute(S, D, mlist)**Input:** S, D [S=Source, D=Destinationt]**Output:** mlist1.***Begin***2.*mlist=S*3.*path=false*4.*nlist=FindNeighbour(S)*5.***While*** *(nlist!=phi)*6.*do*7.   ***For*** *each Vi* ∈ *nlist*
8.*do*9.***If*** *Vi=D then*10.  *path=true*
11. *mlist=mlist*
 ∪ 
*Vi*
12.  *break;*
13. ***End If***
14.***If*** *Vi* ∉ *mlist **then***15.*temp=FindNeighbour(Vi)*16.*nlist=append(nlist, temp)*17.*mlist=mlist U Vi*18.***End If***19.***End For***20.***If*** *path=true then*21.*break;*22.***End If***23.***End while***24.***If*** *path=true then*25.*Return mlist*26.***End If***27.***else***28.*Return null*29.***End***

As stated in Algorithm 3, the *FindNeighbour* function determines the nest hop. To calculate the next hop, the whole range of the node was divided into four different zones, as illustrated in Figure 12.

The nodes available near the sender node will be available for longer, but it will increase the hop count. If the next hop is available inside *PZ1* (Priority Zone 1), the hop count will be lower. If a node is not available in *PZ1*, the next priority is given to *PZ2*, then *PZ3*, and so on. The division of the zones is decided based on the simulation results.
**Algorithm 3**: FindNeighbour**Input**: VID=Vehicle ID**Output**: nexthop1.***Begin***2.*Priority:=0*3.
   ***for each** Vn in Vehicle_List[n][0…n]*
4.
       ***If** Vn.Priority ≥ Priority **then***
5.
           *Priority=Vn.Priority*
6.
        *Temnode.loc=Vn.loc;*
7.
        *Tempnode.id=Vn.VID*
8.
        *Tempnode.RSU_ID=Vn.RSU-ID*
9.
           ***End If***
10.
      ***End For***
11.
      ***Reutrn** Tempnode*
12.
 ***End***


### 5.3. Packet Forwarding

Once the route is decided, the route is sent to the nodes, which are involved in forwarding packets. The source node starts sending data with the route. Each hop in the route is aware of the next hop, as the RSU sends the route to all nodes. If the route link breaks down in the middle of forwarding data, the hop for which the next hop is lost sends a message to the RSU, and the RSU provides the new route to the nodes. A session is maintained at the sender node. The sender starts sending packets from the end point of the previous session whenever the RSU provides the new route. Algorithm 4 defines the packet forwarding mechanism.
**Algorithm 4**: PacketForward (S, D, H, P)**Input**: S = Source Node, D = Destination Node, H = Next Hop, P = Packet**Output**: Acknowledgement1.***Begin***2.*ACK* ← *sendPacket(S, D, H, P) //Sends the data packet*3.***If***
 *ACK is not found **then***
4.
   *Ctr:=0; //initialise the counter*
   *ACK:=null // initialise the acknowledgement null*
   ***While*** *Ctr<ttl **do***
5.
    *ACK*
 ← 
*SendPacket(S, D, H, P)*
6.
      ***If***
 *ACK is found **then***
7.
      *Return ACK;*
8.
      ***End If***
9.
    ***Done***
10.
    ***If***
 *ACK is null **then***
11.
      *InformRSU(S, D, H)*
12.
    ***End If***
13.
   ***Else***
14.
      *Return ACK;*
15.
   ***End If***
16.
   ***End***


## 6. Computational Complexity of RC-LAHR

The computational complexity of any algorithm defines the viability of the algorithm in the real world. In this section, we compute the complexity of the proposed algorithm. Nomenclature lists some symbols used in the analysis.

**Theorem** **1.***The upper bound of the route discovery with n number of nodes is always O(n^3^)*.

**Proof** **of Theorem 1.**The route discovery overhead can be calculated by the sum of the RREQ overhead and the overhead lay on RREP. Hence, the route discovery overhead can be represented by the following formula.
(1)$$Route\;discovery\;overhead=\prod_{RREQ}+\prod_{RREP}$$The overhead of sending the RREQ packet to the RSU has only the overhead of propagation delay from the node to the RSU. Hence, the overhead can be defined by the following formula.
(2)$$\prod_{RREQ}={\Delta t}_{1}$$The delay of RREP consists of the time of route discovery and the propagation delay of the transmission of RREP to the nodes. The framework finds the route at layer 1, and if the destination is not present at layer 1, it sends the RREQ to layer 2, where the route is calculated and is sent to the RSU, and then the RSU sends the RREP to the nodes. Hence, the delay at route discovery can be defined as follows.
(3)$$\delta_{RREP}=\tau_{Rdiscv}+{\Delta t}_{1}\;\tau_{Rdiscv}=max\left\{\begin{array}{c} D_{Rdiscl1}\\ D_{Rdiscl2} \end{array}\right.$$The route discovery at layer 1, i.e., the RSU, involves the process of finding source and destination, route calculation, propagation delay from the RSU to the nodes, and some other constraints like distraction, etc. The following equation summarizes it.
(4)$$D_{Rdiscl1}=T_{Froute}+\tau_{snoder}+{\Delta t}_{1}+C$$The route discovery at layer 2 takes place when the destination node is not present in the range of the RSU where the RREQ was initiated. Hence, the route discovery at layer 2 involves the node search, route calculation, and propagation delay in addition to the time taken to search for the node at layer 2. The following formula completely describes it. C represents the other constant.
(5)$$D_{Rdiscl2}=\tau_{snoder}+{\Delta t}_{2}+T_{Froute}+\tau_{snodel2}+C$$We use the hashing technique at the RSU to search for the node, hence $\tau_{snoder}$ will take O(1) time. We assume the propagation delay is constant, therefore it takes ${\Delta t}_{2}=$O(1). The route find algo requires to traverse n node’s neighbor, the time taken by the algo i.e., $T_{Froute}=$ O(n^2^). However, at layer 2, we have the global view of the topology, we do not have any constraint of storage, and we can afford the high-speed processor; we used B+ Tree to store the nodes. The time taken to search for the node at layer 2, $\tau_{snodel2}=$ O(nlogn).Hence, the time complexity of $D_{Rdiscl2}$ = O(1) + O(1) + O(n^2^) + O(nlogn) + O(1) = O(n^2^)Likewise, the time complexity of $D_{Rdiscl1}$ = O(n^2^) + O(1) + O(1) + O(1) = O(n^2^)Hence, the routing overhead is O(n^2^) for one route discovery. Let us suppose, in the worst-case scenario, there are *n* nodes, and all these nodes send the RREQ. In this case, the complexity is n* O(n^2^) = O(n^3^).□

## 7. Performance Evaluation and Analysis of Results

In this study, Mininet Wi-Fi [47] along with SUMO [48] was used for the simulation. This model was setup on Ubuntu Server, which runs on a virtual machine. Figure 13a,b illustrate the topology and SUMO map used for the simulation. Table 1 lists the essential parameters used in the simulation.

The performance of the routing protocol was assessed through simulation. This study first investigated the influence of range, PDR, and EED on the proposed protocol. Subsequently, the link’s reliability was examined under different speed conditions, providing valuable insights into its adaptability. Additionally, the analysis included an evaluation of crucial metrics such as PDR, EED, and the handling capability of the traffic density. The PDR is computed as the ratio of the number of packets received to the total number of packets transmitted. The average amount of time that passes between when a packet is sent from its source node and when it arrives at its destination node is referred to as the EED. The proposed routing protocol was also assessed based on traffic density and load on the RSU and cloud for route finding. The proposed routing protocol was compared with GPSR and OPBR.

### 7.1. Packet Delivery Ratio Analysis

Figure 14 depicts the variations in the packet delivery ratio in relation to the speed of the vehicle. It may be deduced from the fact that the speed of the vehicle influences the transfer of data packets. A change in the speed of the vehicle at a specific point in time influences the selection of the packet’s next relay node, which will, in turn, affect the delivery ratio of the packets. The SDN controller in RC-LAHR determines the route. The route determination is performed using the location of the vehicles and the priority zone. Hence, there is less possibility of route disconnection, which leads to a higher PDR. It is observed from Figure 14 that the average PDR of RC-LAHR is better in comparison with the GPSR and OPBR.

The expansion of the communication range led to an increase in the total packet delivery ratio. This may be considered a positive development. Within the context of this simulation scenario, the maximum permissible speed for the vehicles is 40 km/h. There is the possibility of additional nodes being joined because of the vehicle communication range. Figure 15 clearly shows that the PDR is better when the communication range is increased. The analysis of PDR vs. communication range is necessary for the further analysis of the network, like network density and EED.

Figure 16 depicts a range of different packet transfer rates vs. the percentage of packets that are successfully delivered to their intended destinations. When there is a rise in the packet rate, also known as the number of data packets per second, there is a probability that there will also be an increase in the packet delivery ratio initially. However, if the pace at which the packets are sent remains high, the capacity of the network’s communication channels will be depleted, and the proportion of packets that are successfully delivered will either stay the same or decrease. Figure 16 clearly shows that RC-LAHR performs better than the other protocols.

### 7.2. End-to-End Delay Analysis

The end-to-end delay time is the time taken for a data packet to travel from its source to its destination across the network. In RC-LAHR, the next hop is calculated based on the priority zone. The vehicle available in a higher priority zone is selected for the next hop. Hence, the proposed routing approach always selects the route that has the minimum number of hops, which results in a consistently low amount of delay. In Figure 17, the average end-to-end delay is shown against the communication range. In this simulation, the vehicle speed was taken to be 40 km/h. The delay time for this routing method is smaller than the delay time for the other two routing methods. Figure 18 illustrates the effect of vehicle speed on the end-to-end delay. For this, a communication range of 300 m and 10 packets per second as data rates were assumed. Likewise, Figure 19 evaluates the EED of RC-LAHR based on different data packet rates. RC-LAHR gives priority to the nodes that have a higher chance of being in range. Hence, the route selected is more likely to be stable. Therefore, once the route is set up, there is very little delay in packet delivery. It is observed that RC-LAHR performs better in both cases.

### 7.3. Network Traffic Analysis

Network traffic density is a measurement of the quantity of data transmitted or received on a network during a given time interval. The data packets, route requests, and request replies to packets were taken as parameters to analyze the traffic density. Initially, in the proposed routing protocol, the routing packets were sent frequently by all vehicles to the RSU for the initialization process. Because of this, the traffic density in the initial phase took more time. After the establishment of all the vehicles and the RSU’s routing table, the topology of the network was stored on the SDN controller (RSUs). Hence, whenever the route was required by any vehicle, the route was determined immediately and sent to the vehicle. Due to this, the route reply and route request times were reduced. This resulted in a lower average traffic density. In Figure 20, the impact of the traffic density during the simulation is analyzed.

### 7.4. Route Calculation Time Analysis

It is necessary to evaluate the suggested routing protocol according to the amount of time needed to find the route. Initially, the information about the topology was not stored at the SDN controller. Therefore, the route calculation took more time. Once the SDN controller stored information about the overall topology, the route determination time was reduced. Figure 21 compares the amount of time spent route-finding to the amount of time spent simulating, while Figure 22 compares the average amount of time spent to find the route to the total number of connections. The simulation used 30 nodes with the speed of 40 km per hour.

### 7.5. Load Analysis

The proposed routing protocol heavily relies on the use of the RSU and the cloud as major resources. Consequently, conducting an examination of the traffic load on the RSUs and the cloud infrastructure throughout the process of route determination has paramount importance. Figure 23 shows a comparison between the loads that the RSUs and the cloud experienced in relation to the total number of connections.

### 7.6. Network Reliability Analysis

VANETs may experience frequent changes in network topology due to the mobility of vehicles. Network reliability in VANETs involves assessing the likelihood that vehicles within the network are able to maintain connectivity with each other and with roadside infrastructure, even as they move through the network. RC-LAHR selects the node that has the highest probability of staying in the network and has a smaller number of hops. To select such hops, RC-LAHR uses the priority zone as discussed in Section 4.1. Figure 24 shows the result of network reliability with the number of connections.

## 8. Discussion

In the dynamic environment of VANETs, where vehicles are frequently moving and network topology changes rapidly, maintaining a high PDR is challenging. The result shows that the PDR has increased significantly. This implies that the routing protocol is highly effective in adapting to these changes, ensuring reliable communication. In VANETs, the timely delivery of information is critical, especially for applications requiring real-time or near-real-time data, like traffic congestion alerts and cooperative driving. The result shows that EED was reduced as compared to the state-of-the-art protocols. The reduced EED can support time-sensitive applications more effectively, enhancing the overall safety and efficiency of vehicular networks. The proposed architecture uses cloud computing and RSUs as an SDN control plane. These SDN control planes provide routing information to the node. The load on the RSUs and on the cloud significantly impacts the proposed routing protocol. The results conclude that the load on the RSUs and the cloud increased as the number of connections increased. The calculation of the time required for routing shows the efficiency and efficacy of the routing protocol.

## 9. Conclusions

In this study, a hybrid hierarchical architecture of SDN was proposed. The proposed architecture perfectly copes with problems like scalability and heterogeneity in VANETs. In this architecture, the control plane was divided into two parts: control plane layer 1 and control plane layer 2. Control plane layer 1 contains RSU-level information about the vehicle, while control plane layer 2 keeps information about the global network. To extend the network, the RSU was placed in the middle of the road, on a divider, to obtain maximum coverage. Further, cloud storage was used for control plane layer 2. A cloud-enabled RSU-based location-aware routing protocol, RC-LAHR, was proposed for the architecture in question. The time duration of the vehicle within the communication range was the selection criterion of the next hop in the route for the routing protocol. The proposed routing protocol, based on the proposed architecture, was compared with existing GPSR and OPBR protocols on different parameters like network reliability, traffic density, and route calculation time. The efficiency and efficacy of the proposed protocol were also tested on PDR and EED. The proposed routing protocol performed well as compared to GPSR and OPBR. The result shows that the EED was reduced to 20% and the PDR was increased to 30%. The network reliability also increased by up to 5% as compared to the OPBR and GPSR.

## 10. Limitations and Future Scope

Despite the promising outcomes of this study, it is important to acknowledge its limitations, which pave the way for future research directions. A major constraint of the research is the increasing load on the RSUs as connections increase. In high-density traffic conditions, this may cause bottlenecks and reduce network efficiency. Cloud storage for control plane layer 2 is advantageous for data processing; however, latency challenges arise when cloud servers are far from the RSUs.

RSU load balancing methods should be improved in future studies. Real-time traffic data might be used to dynamically allocate resources or integrate edge computing to spread the processing burden. Energy constraints are a major challenge, especially for electric vehicles. Therefore, future research should also consider the energy efficiency of the network, especially focusing on the power consumption of the vehicle, RSUs, and cloud servers, aiming to develop more sustainable VANET architectures.

## Figures and Tables

**Figure 1 sensors-24-01045-f001:**
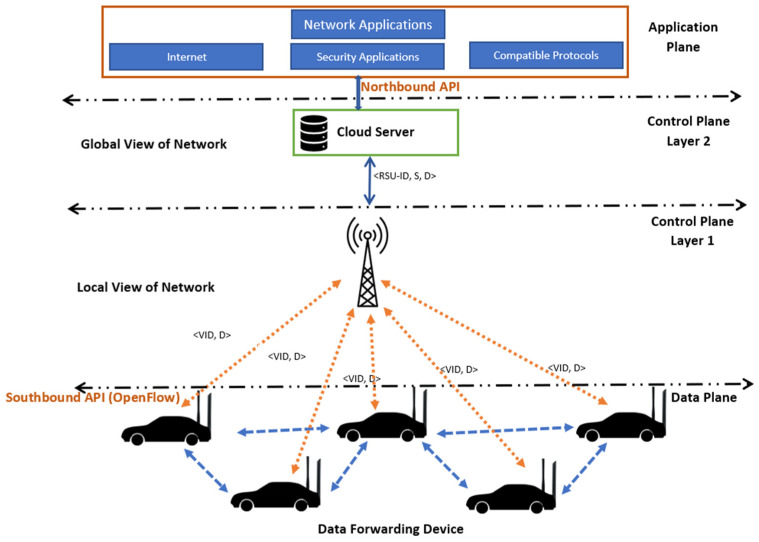
An illustration of the system model with the infrastructure for a hierarchical SD-VANET.

**Figure 2 sensors-24-01045-f002:**
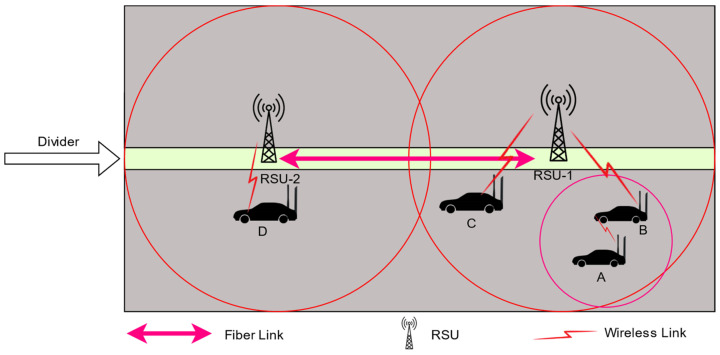
Routing scenarios. A is the sender node and B, C, D are the possible destination nodes. According to these possible nodes, the following four types of scenarios are possible.

**Figure 3 sensors-24-01045-f003:**
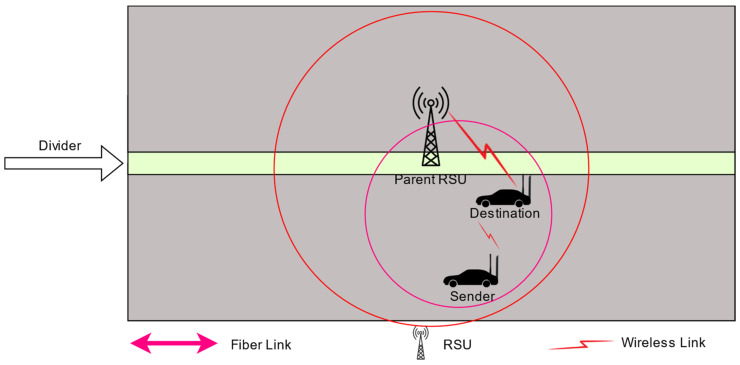
The destination node is inside the range of the sender node.

**Figure 4 sensors-24-01045-f004:**
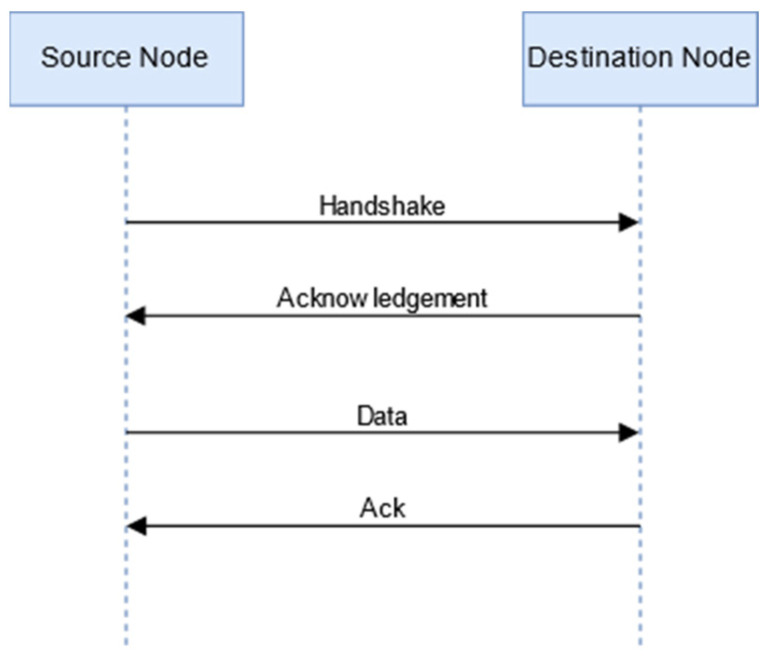
Communication between source node and destination node when the destination node is found in the range of the sender vehicle.

**Figure 5 sensors-24-01045-f005:**
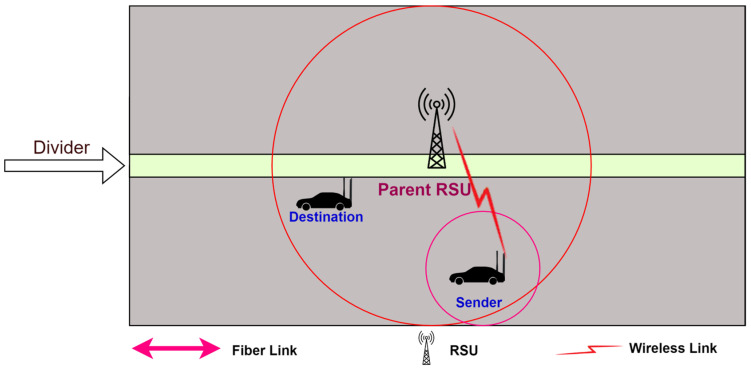
The destination node is inside the range of the parent RSU but outside the range of the sender node.

**Figure 6 sensors-24-01045-f006:**
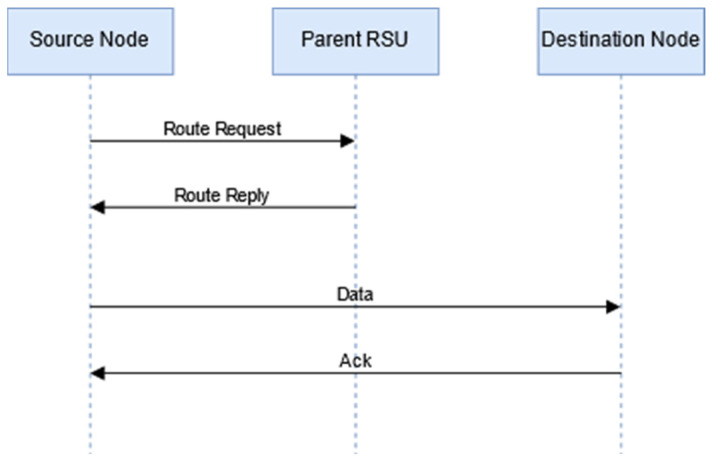
Communication between source node and destination node when the destination node is found in the range of the RSU but outside the range of the sender.

**Figure 7 sensors-24-01045-f007:**
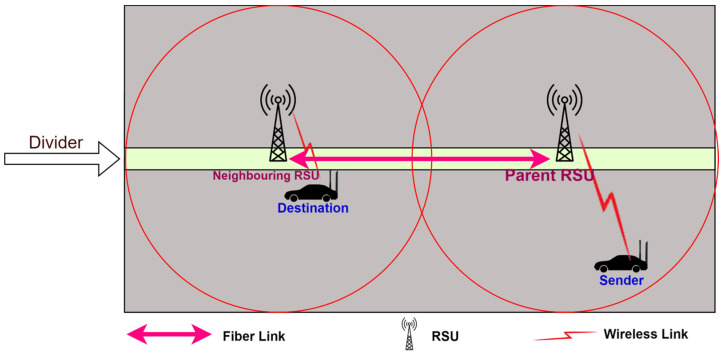
The destination node is outside the range of the parent RSU but inside the range of a neighboring RSU.

**Figure 8 sensors-24-01045-f008:**
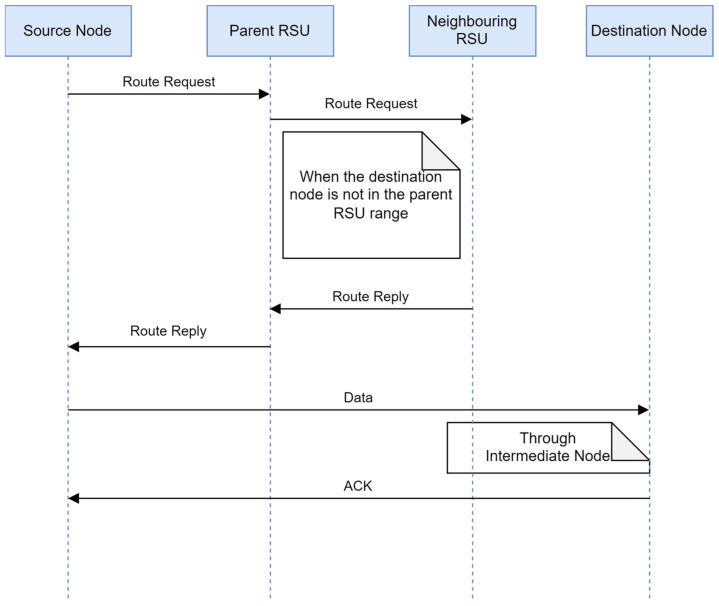
Communication between source node and destination node when the destination node is outside the range of the RSU but in the range of a neighboring RSU.

**Figure 9 sensors-24-01045-f009:**
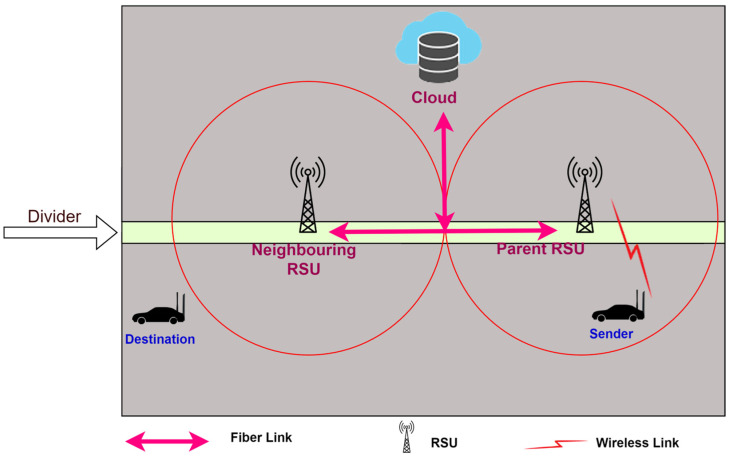
The destination node is outside the range of neighboring RSUs.

**Figure 10 sensors-24-01045-f010:**
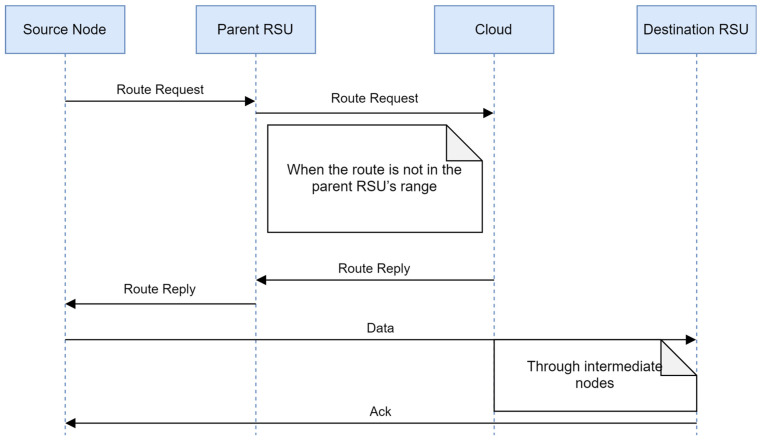
Communication between source node and destination node when the destination is not in the range of neighboring RSUs.

**Figure 11 sensors-24-01045-f011:**
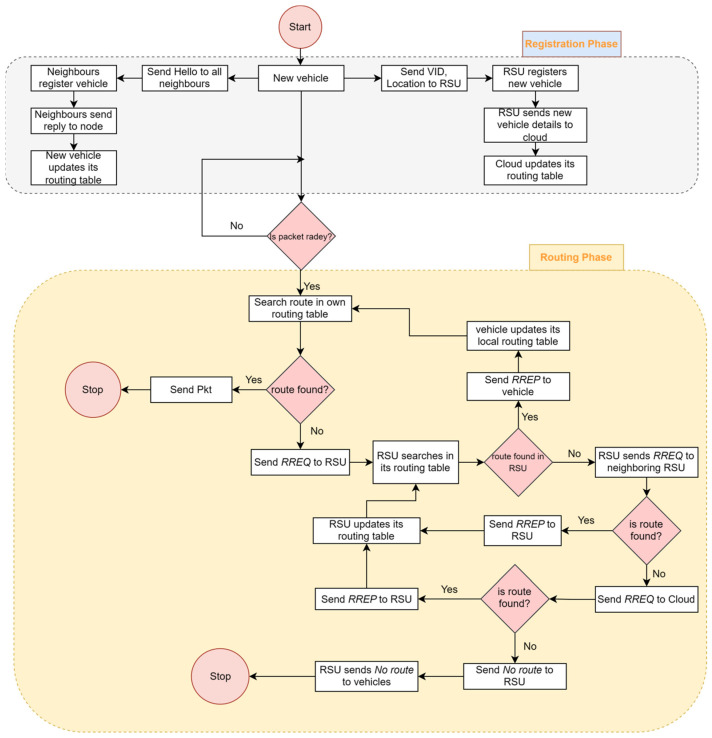
Working flow of proposed routing protocol.

**Figure 12 sensors-24-01045-f012:**
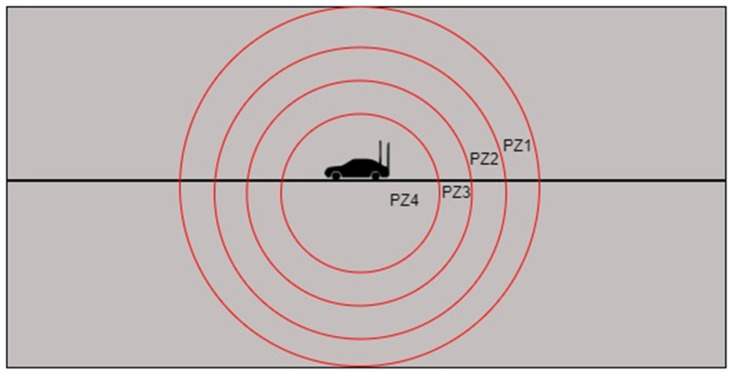
Priority zone division for next hop calculation.

**Figure 13 sensors-24-01045-f013:**
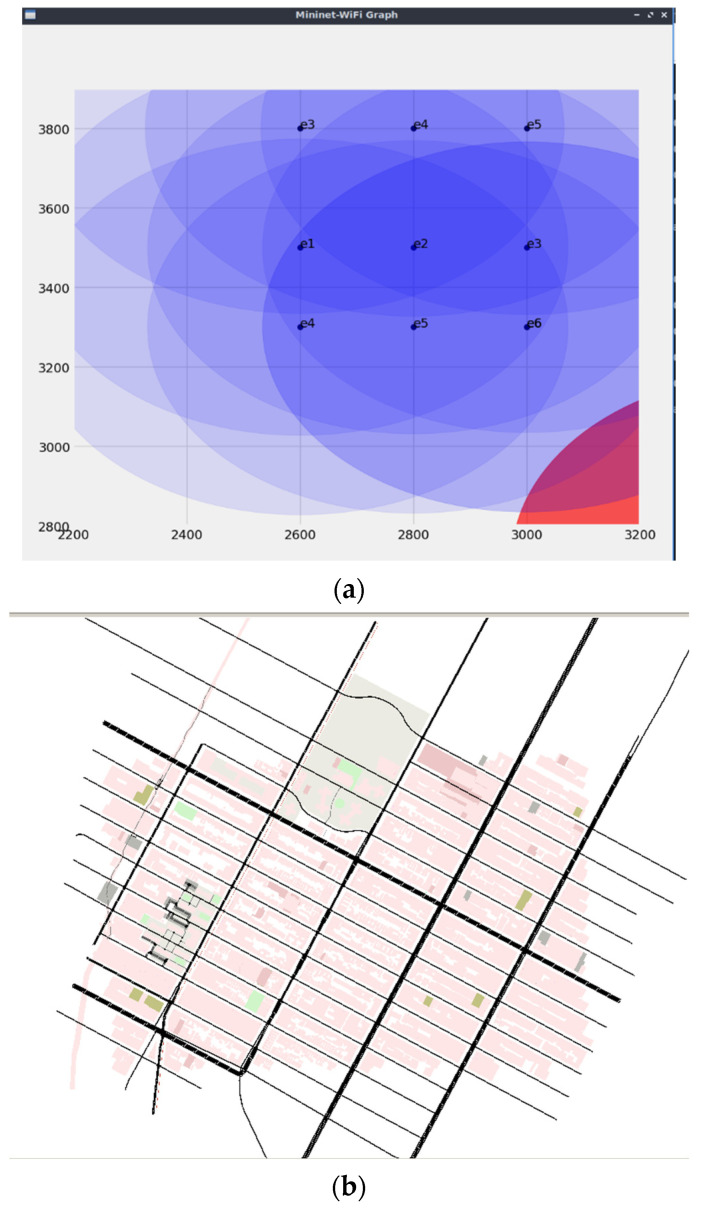
(**a**) Mininet Wi-Fi topology used for the simulation. (**b**) SUMO map used for the simulation.

**Figure 14 sensors-24-01045-f014:**
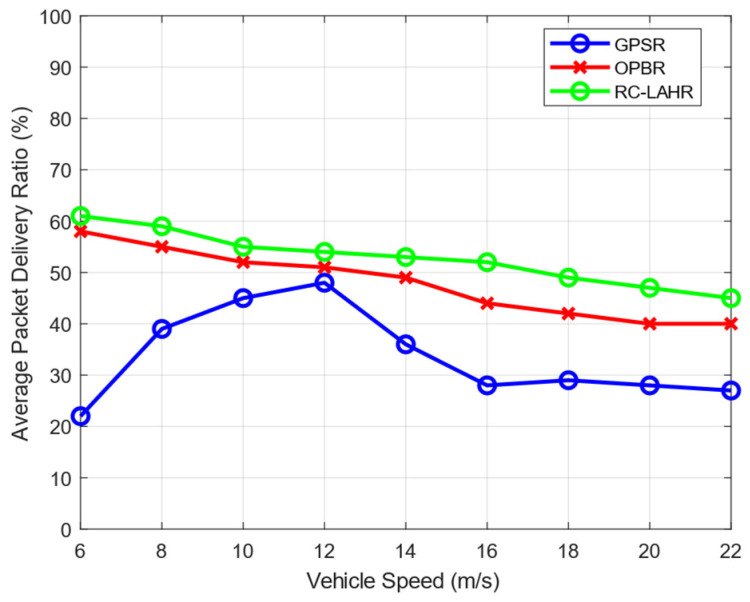
Average packet delivery ratio vs. vehicle speed.

**Figure 15 sensors-24-01045-f015:**
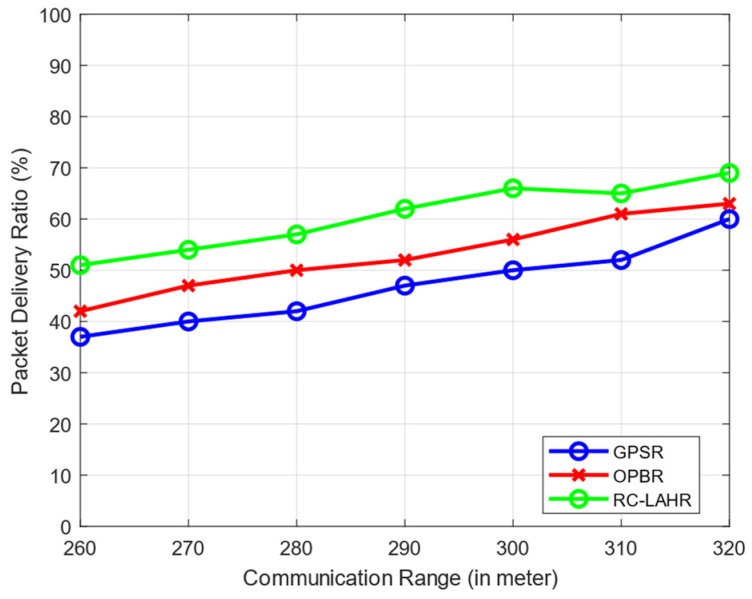
Packet delivery ratio vs. communication range.

**Figure 16 sensors-24-01045-f016:**
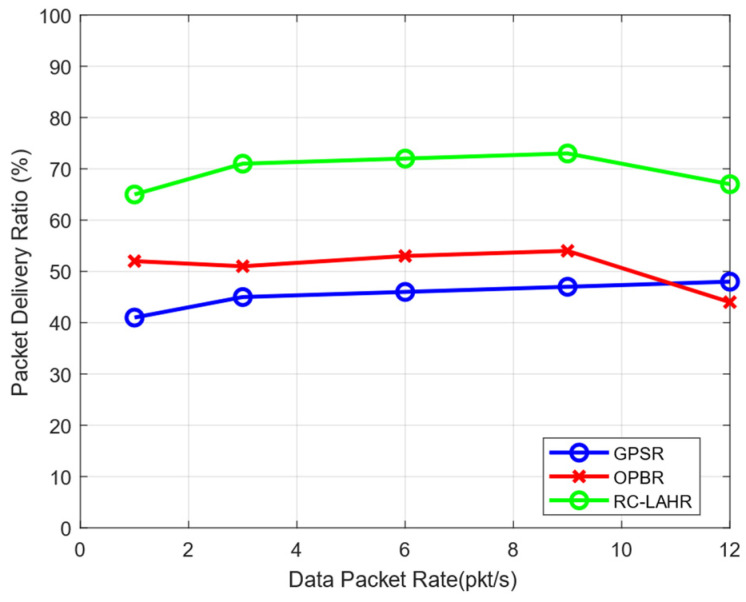
Packet delivery ratio vs. data packet rate.

**Figure 17 sensors-24-01045-f017:**
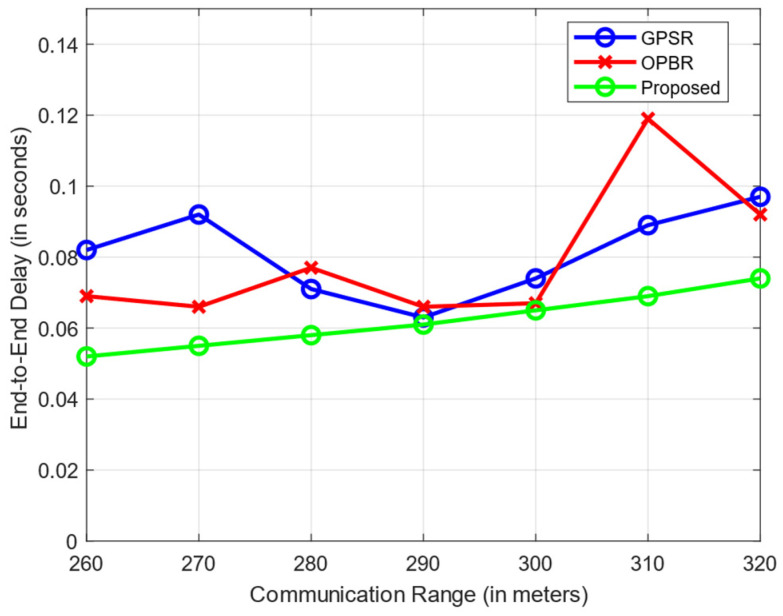
End-to-end delay vs. communication range.

**Figure 18 sensors-24-01045-f018:**
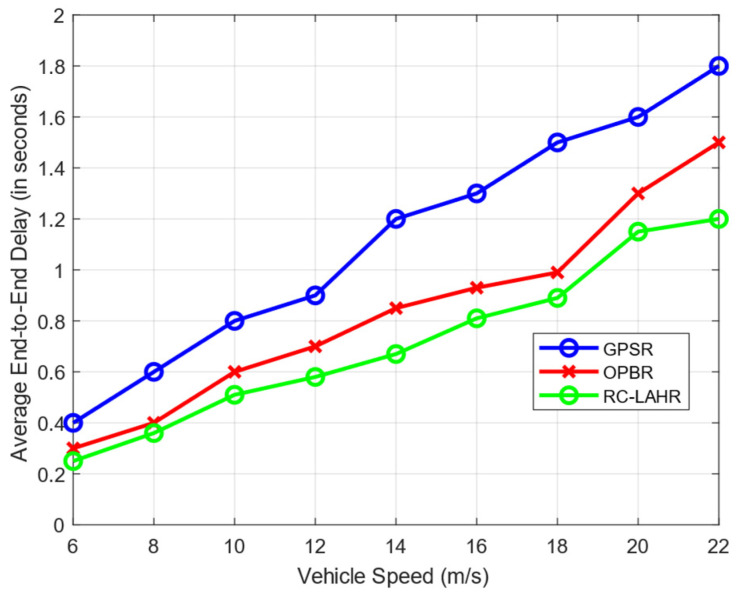
End-to-end delay vs. vehicle speed.

**Figure 19 sensors-24-01045-f019:**
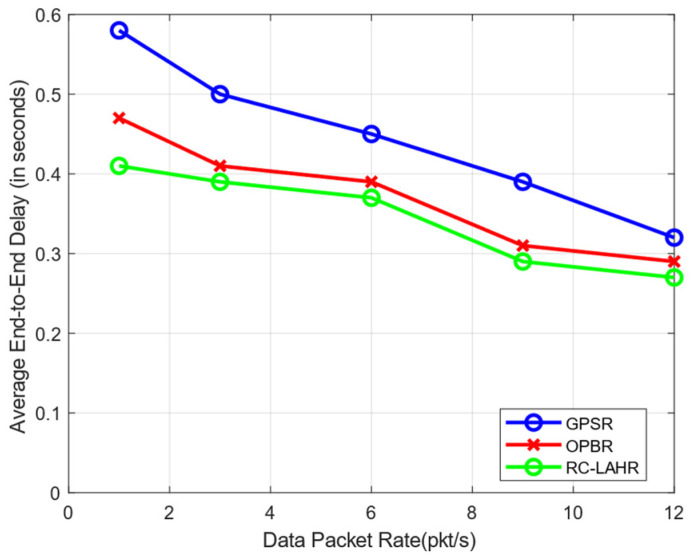
End-to-end delay vs. data packet rate.

**Figure 20 sensors-24-01045-f020:**
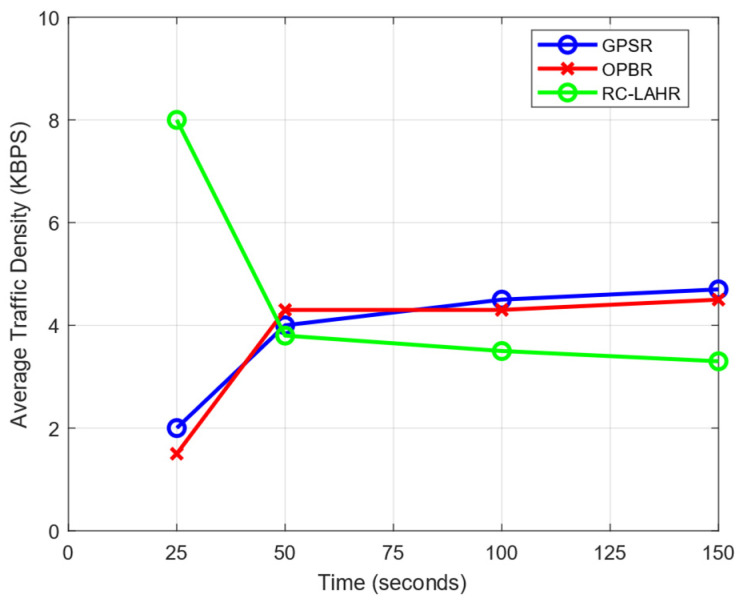
Average traffic density vs. time.

**Figure 21 sensors-24-01045-f021:**
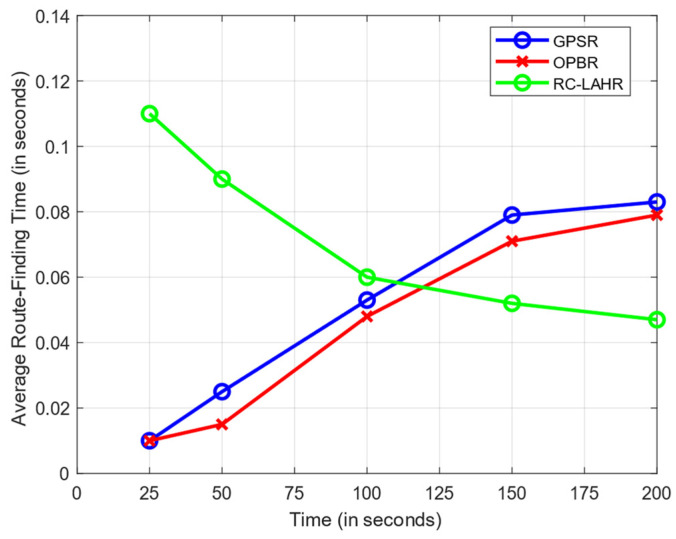
Average route calculation time vs. time.

**Figure 22 sensors-24-01045-f022:**
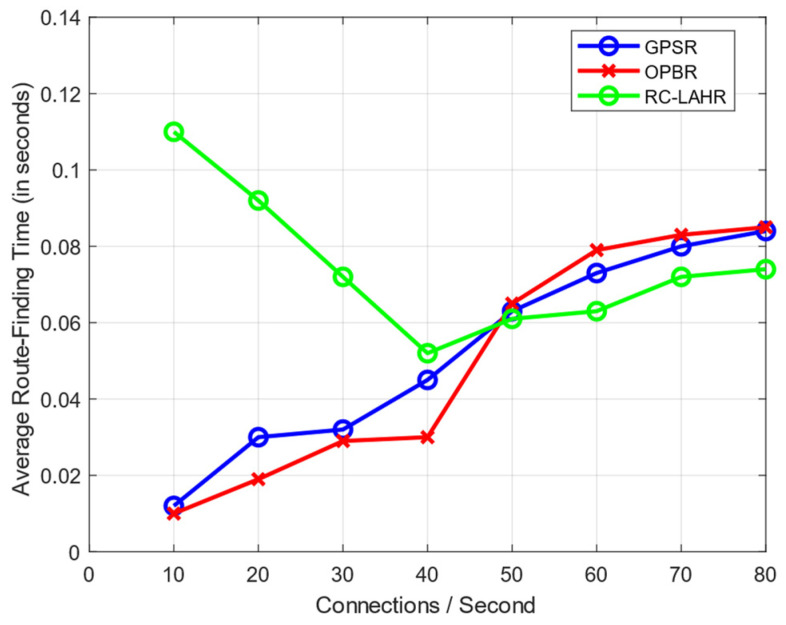
Average route-finding time vs. connection rate.

**Figure 23 sensors-24-01045-f023:**
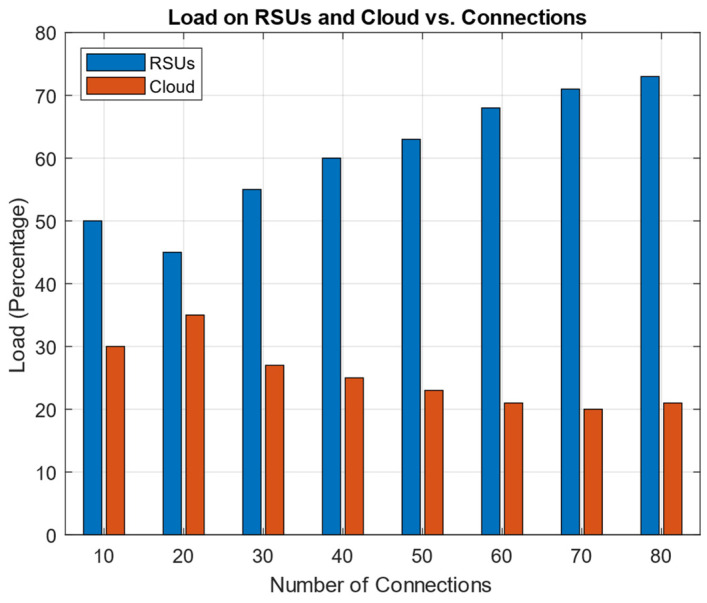
Load on RSUs and cloud vs. number of connections.

**Figure 24 sensors-24-01045-f024:**
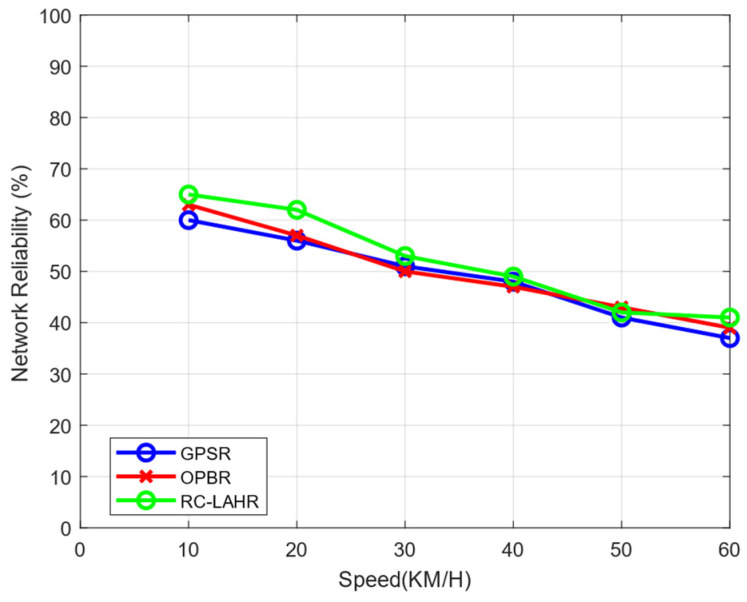
Network reliability vs. speed.

**Table 1 sensors-24-01045-t001:** Simulation parameter settings.

Parameters	Setting Value
Simulated environment size	3800 m × 3200 m
MAC protocol	IEEE 802.11p
Transmission range	260–320 m
Transmission rate range	1–12 Mbps
Controller	OpenFlow
Speed range	20–80 km/h
Hello packet cycle	50 ms
Simulation time	200 s
Number of nodes	50 nodes

## Data Availability

No new data were created or analyzed in this study. Data sharing is not applicable to this article.

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
