# Peer review of "RC-LAHR: Road-Side-Unit-Assisted Cloud-Based Location-Aware Hybrid Routing for Software-Defined Vehicular Ad Hoc Networks"

_sensors, 2024, doi:10.3390/s24041045_

Round 1
Reviewer 1 Report
Comments and Suggestions for Authors
-The research contribution is not clear.
-The abstract should include Aim, Methodology and Results.
-The abstract is unclear.
-in DISCUSSIONS explain how to be linked with RESULTS.
Discussion: More details and discussion are needed in the discussion section.
-References: It is necessary to add a set of references because it is not enough
Al-Rweis, A., zakaraya, Z., Al-Omari, L., & Abdul-Aziz , K. (2024). Impact of smoking on Galectin-3 and GDF-15 among pregnant women. Tamjeed Journal of Healthcare Engineering and Science Technology, 2(1), 1–12. https://doi.org/10.59785/tjhest.v2i1.37
JASIM, F. T., & M., K. (2023). ARTIFICIAL INTELLIGENCE INNOVATION AND HUMAN RESOURCE RECRUITMENT. Tamjeed Journal of Healthcare Engineering and Science Technology, 1(2), 20–29. https://doi.org/10.59785/tjhest.v1i2.22
Kadhim, N. M., Mohammed, H. A., Radhawi, S. N., Jabur, A. M., Gottraan, R. B., Abdulridha, M. M., Kadhim, W. M., & Mohammed, Z. Q. (2023). Investigation of the next generation science standards including in the science book according to E-learn : analytical study. Tamjeed Journal of Healthcare Engineering and Science Technology, 1(2), 30–35. https://doi.org/10.59785/tjhest.v1i2.23
Thivagar, M. L., Al-Obeidi, A. S., Tamilarasan, B., & Hamad, A. A. (2022). Dynamic analysis and projective synchronization of a new 4D system. In IoT and Analytics for Sensor Networks: Proceedings of ICWSNUCA 2021 (pp. 323-332). Springer Singapore.
Alsaffar, M., Hamad, A. A., Alshammari, A., Alshammari, G., Almurayziq, T. S., Mohammed, M. S., & Enbeyle, W. (2021). Network management system for IoT based on dynamic systems. Computational and Mathematical Methods in Medicine, 2021.
Comments on the Quality of English Language
-The research contribution is not clear.
-The abstract should include Aim, Methodology and Results.
-The abstract is unclear.
-in DISCUSSIONS explain how to be linked with RESULTS.
Discussion: More details and discussion are needed in the discussion section.
-References: It is necessary to add a set of references because it is not enough
Al-Rweis, A., zakaraya, Z., Al-Omari, L., & Abdul-Aziz , K. (2024). Impact of smoking on Galectin-3 and GDF-15 among pregnant women. Tamjeed Journal of Healthcare Engineering and Science Technology, 2(1), 1–12. https://doi.org/10.59785/tjhest.v2i1.37
JASIM, F. T., & M., K. (2023). ARTIFICIAL INTELLIGENCE INNOVATION AND HUMAN RESOURCE RECRUITMENT. Tamjeed Journal of Healthcare Engineering and Science Technology, 1(2), 20–29. https://doi.org/10.59785/tjhest.v1i2.22
Kadhim, N. M., Mohammed, H. A., Radhawi, S. N., Jabur, A. M., Gottraan, R. B., Abdulridha, M. M., Kadhim, W. M., & Mohammed, Z. Q. (2023). Investigation of the next generation science standards including in the science book according to E-learn : analytical study. Tamjeed Journal of Healthcare Engineering and Science Technology, 1(2), 30–35. https://doi.org/10.59785/tjhest.v1i2.23
Thivagar, M. L., Al-Obeidi, A. S., Tamilarasan, B., & Hamad, A. A. (2022). Dynamic analysis and projective synchronization of a new 4D system. In IoT and Analytics for Sensor Networks: Proceedings of ICWSNUCA 2021 (pp. 323-332). Springer Singapore.
Alsaffar, M., Hamad, A. A., Alshammari, A., Alshammari, G., Almurayziq, T. S., Mohammed, M. S., & Enbeyle, W. (2021). Network management system for IoT based on dynamic systems. Computational and Mathematical Methods in Medicine, 2021.
Author Response
The authors are grateful to the respected reviewer for the valuable comments. The point-to-point response of the reviewer is attached herewith. Please see the attachment.
With Regards

Reviewer 2 Report
Comments and Suggestions for Authors
The research in this work has the focus on securing the communication link in VANETs which is very much required for secure communication.
The article is well written.
However, there are few minor corrections which can be addressed:
1. Abbrevations must be declared only once and there is no need to re-declare them again in the paper. Instead, abbreviations then should be used through out. Few abbreviations are re-declared many times, and few abbreviations like OBU mentioned in the abbreviation table has not been used anywhere in the paper. So, abbreviation table should only have those abbreviations which are used in the article.
2. Keywords must be in alphabetical order.
3. Section 3 Motivation and Problem Definition is very short and must be merged with section 1. No need of creating a seperate section of one paragraph.
4. Page 16, line 387, "python is used to write the script and to show the image". Which image author is talking about and which script? Explain. If there is some code, provide the description of that and upload that on some repo.
5. How many nodes are considered? and also compare the work with some existing work in literature.
Author Response

(The authors gave the same response as above.)

Reviewer 3 Report
Comments and Suggestions for Authors
Authors have proposed RC-LAHR: RSU Assisted Cloud based Location Aware Hybrid Routing for Software Defined VANET. However, author need to address the following queries:
1. The abstract should always mention the rate of efficacy/efficiency percentage of the proposed method for the reader’s quick overview.
2. The abstract should at least have a line or two about the need for this work. This abstract has an intro and it straightaway deals with the proposed work.
3. In the introduction section author needs to provide their major contribution. Moreover, the separate section for research gap, objective and motivation needs to be defined in a clear manner.
4. Literature section is very weak authors should specify techniques, advantages, and limitation of the existing approaches. Moreover, at the end of the survey authors should specify how their proposed approach overcomes the limitation or gap of the existing protocols in terms of routing
5. The mathematical justification of the proposed algorithm is missing. That is a mandate for any newly proposed technique.
6. Symbol description table should be included. Due to the absence, it’s very hard to follow up the equations.
7. Lacks in formal analysis of the complexity of relevant solutions.
8. Unfortunately, the language and sentence structures of this manuscript are at times incomprehensible. The paper needs rewriting and thorough language editing to allow for a proper peer review.
9 Rational choice of methods. Explain your own improvement and advantages after your own progress.
10. Authors are advised to follow the IMRAD format for the entire paper.
11. A separate section for Limitations and future work in detail would give further ideas for the readers who wish to enhance your work.
12. The results sections are very weak. authors should clearly specify how proposed system performs better than any other existing system.
13. Authors may include the following references if they are relevant:
a.) An elliptic curve cryptography-based certificate-less signature aggregation scheme for efficient authentication in vehicular ad hoc networks
b.) LAPEP—Lightweight Authentication Protocol with Enhanced Privacy for effective secured communication in vehicular ad-hoc network
c.) Energy-efficient trust-aware secured neuro-fuzzy clustering with sparrow search optimization in wireless sensor network
d.) A comprehensive survey on fuzzy-based intelligent intrusion detection system for internet of things
e.) Energy efficient secured K means based unequal fuzzy clustering algorithm for efficient reprogramming in wireless sensor networks
f.) Intelligent fuzzy rule-based approach with outlier detection for secured routing in WSN
g.) An energy efficient clustered gravitational and fuzzy based routing algorithm in WSNs
Comments on the Quality of English LanguageUnfortunately, the language and sentence structures of this manuscript are at times incomprehensible. The paper needs rewriting and thorough language editing to allow for a proper peer review.
Author Response

(The authors gave the same response as above.)

Round 2
Reviewer 3 Report
Comments and Suggestions for Authors
Authors have addressed all my concerns. Hence the manuscript is accepted